# Developmental hourglass: Verification by numerical evolution and elucidation by dynamical-systems theory

**Takahiro Kohsokabe**[1], **Shigeru Kuratanai**[1], **Kunihiko Kaneko**[2]*

**1** Center for Biosystems Dynamics Research, RIKEN, Kobe, Japan, **2** Niels Bohr Institute, University of Copenhagen, Copenhagen, Denmark

* kunihiko.kaneko@nbi.ku.dk

**Data Availability Statement:** All programs that are needed to reproduce the results in this manuscript are available on Zenodo at https://doi.org/10.5281/zenodo.8049347

## Abstract

Determining the general laws between evolution and development is a fundamental biological challenge. Developmental hourglasses have attracted increased attention as candidates for such laws, but the necessity of their emergence remains elusive. We conducted evolutionary simulations of developmental processes to confirm the emergence of the developmental hourglass and unveiled its establishment. We considered organisms consisting of cells containing identical gene networks that control morphogenesis and evolved them under selection pressure to induce more cell types. By computing the similarity between the spatial patterns of gene expression of two species that evolved from a common ancestor, a developmental hourglass was observed, that is, there was a correlation peak in the intermediate stage of development. The fraction of pleiotropic genes increased, whereas the variance in individuals decreased, consistent with previous experimental reports. Reduction of the unavoidable variance by initial or developmental noise, essential for survival, was achieved up to the hourglass bottleneck stage, followed by diversification in developmental processes, whose timing is controlled by the slow expression dynamics conserved among organisms sharing the hourglass. This study suggests why developmental hourglasses are observed within a certain phylogenetic range of species.

## Author summary

Understanding the intriguing relationship between development and evolution in multi-cellular organisms has long been a challenge in biology. A recent hypothesis called the developmental hourglass proposes that there is a conserved middle stage during development across species of the same animal group. Despite growing evidence supporting this hypothesis, the underlying mechanisms and reasons for its emergence have remained elusive due to limited experimental data.

To address this gap, we employed numerical evolution of gene regulation networks controlling pattern formation. Remarkably, our simulations revealed that species that diverged relatively recently in phylogeny displayed the highest similarity during the middle stage of development, which gradually diminished as they diverged further

**Funding:** SK and KK are supported by a Grant-in-Aid for Scientific Research on Innovative Areas (17H06386) from the Ministry of Education, Culture, Sports, Science and Technology (MEXT) of Japan. https://kaken.nii.ac.jp/ja/grant/KAKENHI-PLANNED-17H06386/ " KK is also supported by a Grant-in-Aid for Scientific Research (A)(20H00123) from the Ministry of Education, Culture, Sports, Science and Technology (MEXT) of Japan. https://kaken.nii.ac.jp/ja/grant/KAKENHI-PROJECT-20H00123/ and Novo Nordisk Foundation (0065542). https://novonordiskfonden.dk/en/ The funders had no role in study design, data collection and analysis, decision to publish, or preparation of the manuscript.

**Competing interests:** The authors have declared that no competing interests exist.

phylogenetically. Our findings satisfied not only the criteria of the developmental hourglass but also confirmed several essential characteristics of the developmental hourglass reported in recent experiments. Through theoretical analysis, we further demonstrated that the emergence of the developmental hourglass could be attributed to the acquisition of genes that change slowly and govern developmental processes, which also foster the robustness of development.

By integrating computational simulations, theoretical insights, and previous experimental evidence, our study thus provides a comprehensive understanding of the developmental hourglass, which will unravel the intricate relationship between development and evolution.

## Introduction

Elucidating the possible relationships between evolution and developmental processes is a fundamentally unsolved issue in biology. Nothing to speak of those early works of 19th-century embryologists and evolutionists such as von Baer [1], Darwin [2], and Haeckel [3]; even today, the search for the general laws of the evolution of developmental processes remains one of the most important challenges in the field of evolutionary developmental biology. As a candidate for such a general law, the "developmental hourglass" model has gathered increased attention. According to the developmental hourglass model, multicellular species belonging to the same phylum have the highest embryonic similarity at a particular intermediate period of their development, when the morphological elements characterizing the phylum are formed.

The validity of the developmental hourglass model was first noted based on the coincidence between the phylotypic stage of vertebrates and the period of temporal and spatial progression of the hox genes [4–6] and has been quantitatively verified for various taxa by bioinformatics analyses over the past two decades (for chordates [7–12]; for echinoderms [13–15]; for arthropods [16,17]; for nematodes [18,19]; for mollusks [20] and also suggested beyond the animal kingdom, for flowering plants [21]; for fungus [22]). One may state that the observed data support a developmental hourglass, at least in some phyla.

Despite the growing evidence for the developmental hourglass model, whether it is just a contingency in the evolution of multicellular organisms or a necessity therein, is still in question. In other words, the possible mechanisms underlying the evolutionary acquisition of mid-embryonic conservation remain elusive. Recent experimental studies have concluded that the developmental hourglass is not sufficiently explained by fragility in the bottleneck stage [23,24], but rather, developmental constraints or limitations on phenotypic variability are suggested to be the main driving force of the emergence of the developmental hourglass [23,25]. The relevance of the self-organization processes to the formation of the hourglass was discussed (as well as the origin of the life cycle of multicellular organisms [26]), whereas there is no consensus on the mechanism for the developmental hourglass.

Generally, it is challenging to solve such fundamental issues in the evolution of developmental processes by means of experimentally available data, and the search for mechanisms of the developmental hourglass is no exception. As another approach to this challenge, simulation studies on the evolution of developmental processes have been conducted over the past 20 years. In this case, one can repeat virtual evolution while retaining all the information, such as gene regulatory networks and developmental dynamics. Hypotheses about the possible relationships between development and evolution can then be tested. Theoretical hypotheses have been proposed using this approach regarding the structure of gene regulatory networks that induce segmentation [27–29], testing plausible evolutionary scenarios of segmentation

mechanisms [30,31], the relationship between developmental and mutational robustness [32], and congruence in the developmental and evolutionary processes in both RNA [33] and gene expression patterns [34].

In this study, we conducted an evolutionary simulation to investigate the mechanism behind the establishment of a developmental hourglass. For this purpose, we adopted a dynamical-system scheme to generate spatial patterns, which is commonly adopted in theoretical studies [27,29–31,34]. The model for the simulation can represent cellular states that are determined by gene expression patterns governed by gene regulatory networks and spatial cell-cell interactions, leading to the differentiation of cell types and pattern formation. Individuals must be selected according to a biologically relevant fitness function. We assumed that if there are more cell types within an individual, they attain more functions such that the fitness is higher. This fitness function reflects the idea that different cell types have different cellular functions so that species with many cell types can deal with multiple tasks; hence, they are more adaptive than those with fewer cell types. Each cell type was defined according to the on or off state of the given output genes (see Fig 1B for a schematic representation). In other words, our simulation mimics competition for survival, where individuals acquiring as many cell types as possible are the most adaptive and would leave more offspring.

In the present paper, we use the term "species", not rigorously following its biological definition. It just refers to organisms with different networks (i.e., distinct gene sets), which are branched by earlier generations in the evolution. This is a bit sloppy in terms of the definition of species in biology. Still, we need to distinguish individual organisms developed by different networks from those developed by the identical gene network (i.e. clones under noise) for the following analysis. Accordingly, we need a term distinguishable from just the individuals. Hence, we use the term "species" in the above sense, when needed.

To study the developmental hourglass, the developmental processes of extant species that diverged from a common ancestor must be compared. Thus, the model must represent not only a mother-to-daughter single-chain pedigree but also be able to examine a phylogenetic tree that contains multiple pedigrees as branches. To achieve this, a two-stage evolutionary simulation was performed. In the first stage, an evolutionary simulation of one lineage was performed to obtain one species by selection. In the second stage, evolutionary simulations were executed over the same number of generations, using an ancestor in the lineage of the first-step simulation. Using different random numbers for mutation, extant species branching from the common ancestor were obtained (see Fig 1C for a schematic representation).

Using the above setup, we aimed to examine the correlation between gene expression patterns and the development of multiple species that diverged from a common ancestor. The results showed a developmental hourglass in the sense that the peak of such correlations lies in the middle of development. We also confirmed other reported characteristics, including accumulation of expressed pleiotropic genes and reduction of intra-species gene expression variance at the developmental bottleneck stage. The developmental period of the hourglass bottleneck obtained in the simulations was determined by the genes whose expression changed slowly during development, acting as genes to schedule development. The bottleneck serves as a developmental foundation that sufficiently attenuates the effects of the initial noise to prepare for subsequent developmental processes. Based on these results, the validity of the developmental hourglass and possible mechanisms for its emergence were discussed.

## Results

Herein, we report the results of the simulation of a model based on the three policies described in the introduction. We assumed that the expression dynamics of each gene at each cell

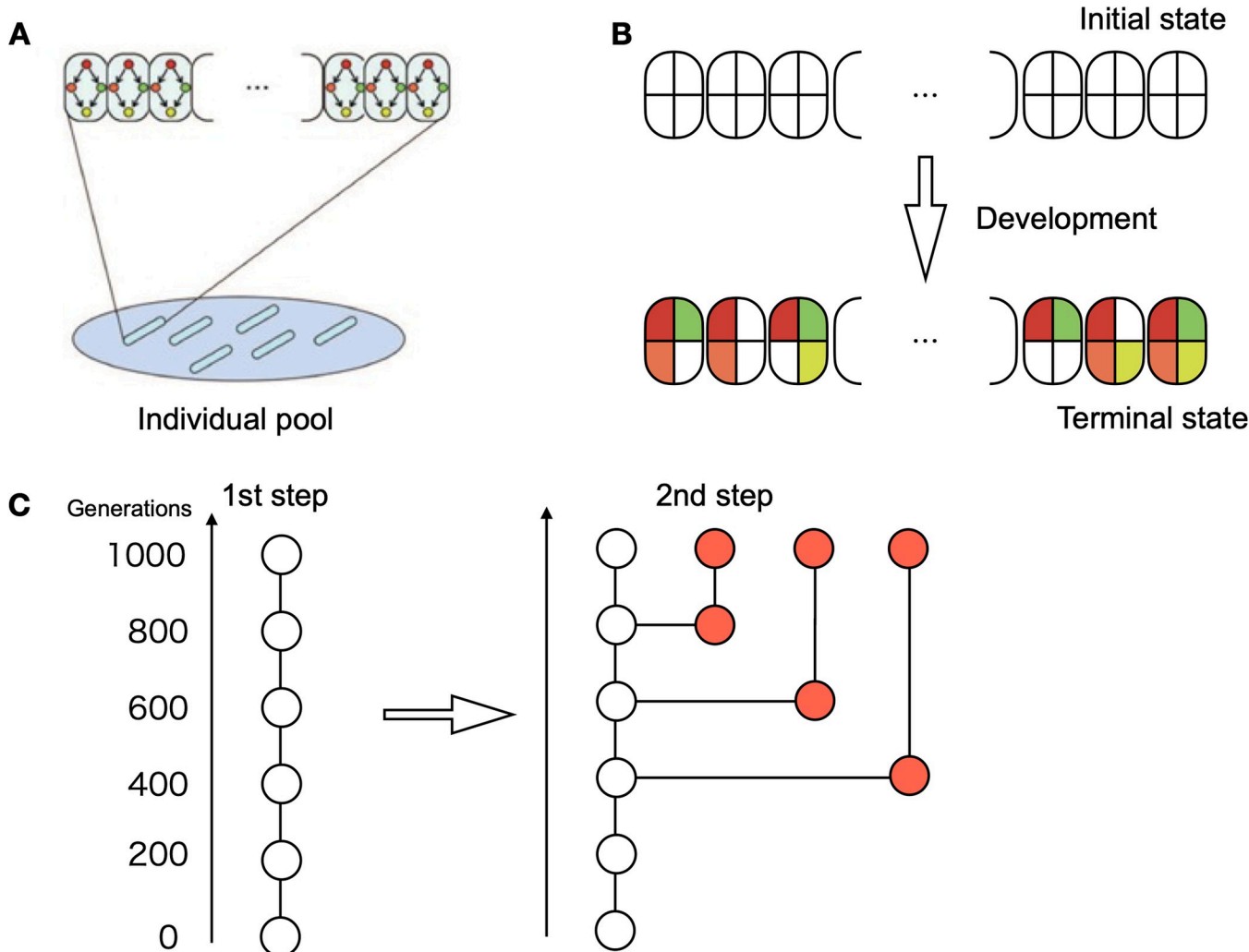

**Fig 1. Schematic of the model used in this simulation.** (A) Each individual, which is a unit of selection, contains cells aligned in a one-dimensional space. These contained the same gene network. The state of each cell, given by the gene expression pattern, changes according to identical gene networks. There were 100 individuals in the evolving population in each generation. (B) Starting from the initial states, the gene expression patterns change according to the gene regulation network. After sufficient time, the expression of each cell reaches the terminal state, on or off. Cell type was determined according to the on/off patterns of the target genes. For the four target genes, there were $2^4$ possible states, as shown in the figure. The fitness of an individual is determined by the number of different cell types in the terminal state. (In the figure, for instance, it is 6). (C) The selection procedure of species diverged from a common ancestor. The left chain represents the result of the first step of evolutionary simulation to select one species. Each circle represents the ancestor at each generation (although significantly abbreviated). As the second step, an ancestor at a certain generation was cloned, and additional evolutionary simulations were carried from the ancestor up to the same generations of the original (1000).

location are governed by the gene regulation network (GRN), while some gene products diffuse through the space. There are two specific morphogens with spatial gradients. These spatial gradients give input to morphogenetic genes. Under those morphogen inputs, the gene expression pattern reached a terminal steady state after sufficient 'developmental' time according to gene expression dynamics governed by the GRN, where the expression of each gene reached an on or off state. Depending on the combination of on/off expression patterns of the given output genes, a cell type is determined (here, we choose four output genes so that a maximum of 16 cell types could exist). Each cell type carries on some functions such that the fitness is given by the number of cell types that coexist over the space in the terminal state. GRNs with higher fitness than others will be selected with a higher probability, whereas slight random

changes are introduced as "mutations" for replication to generate offspring. This simulation mimics evolutionary competition for survival, in which individuals that acquire as many cell types as possible are the most adaptive and leave more offspring. To represent multiple lineages that diverged from a common ancestor, additional evolutionary simulations of ancestral species were performed (see Methods for details).

Fig 2A shows specific examples of the developmental dynamics of two individuals acquired from evolutionary simulation after 1000 generations. The horizontal axis is the developmental

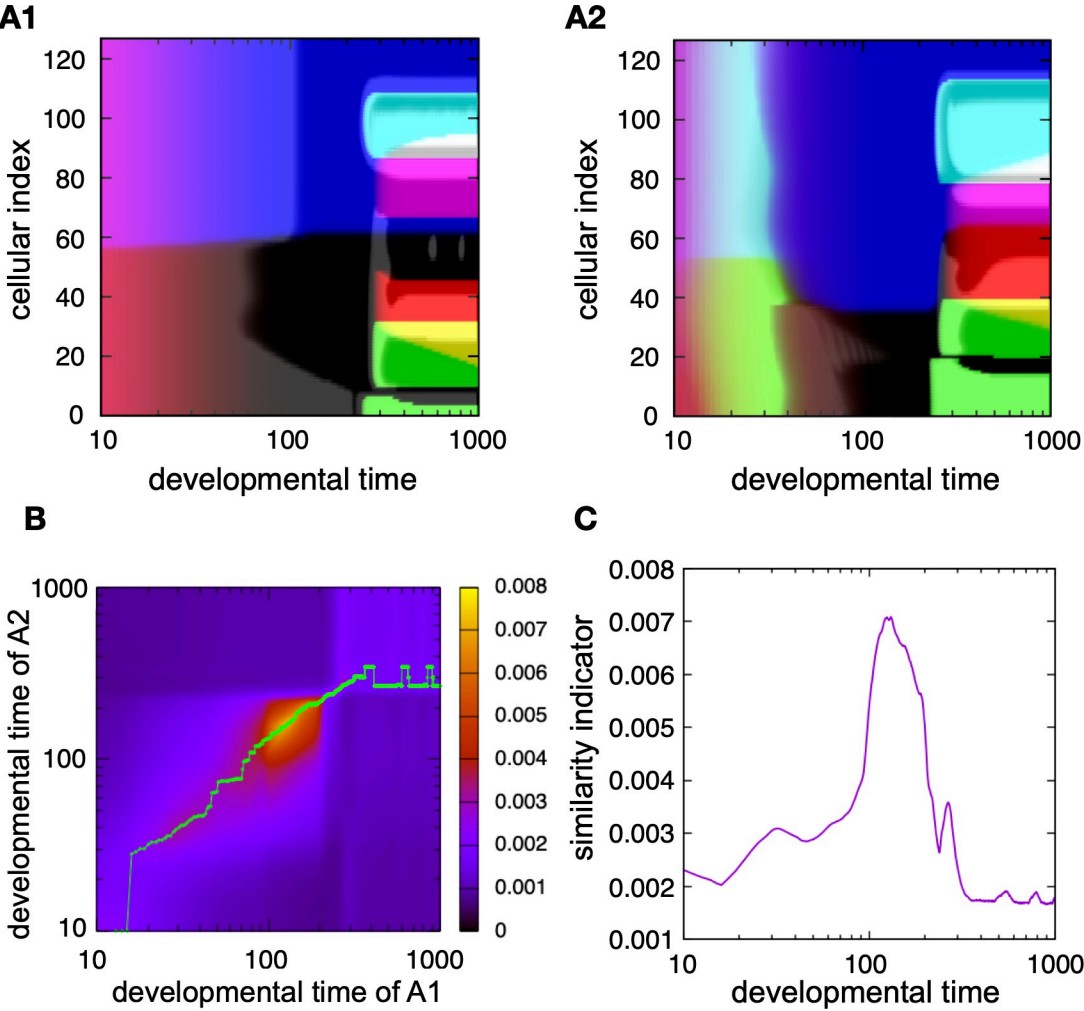

**Fig 2. Examples of the evolutionary simulation that showed hourglass-like developmental correlation.** (A) Two examples of gene expression dynamics of evolved GRNs (A1, A2). The color represents the cellular state, i.e., the expression (On/Off) pattern of four output genes. The vertical axis is the spatial index of cells and the horizontal axis is developmental time. Spatial pattern transitions were more frequent in the early developmental time, and a horizontal log plot was used to adequately depict them. This is consistent with experimental observations. Thereafter, the developmental time was plotted in logs. Following evolving 800 generations, the two GRNs evolved from a common ancestor of the 800th generation and then simulated further over 200 more generations, while suffering from different genetic changes, such that the two have diverged from the common ancestor. (B) The heat map of the similarity indicator, comparing the developmental dynamics of the two individuals given in A. The horizontal axis is the developmental time of the individual A1, and the vertical axis is that of A2. Both axes are on a logarithmic scale. Each point indicates the similarity of gene expression states between the two examples at specific time points. (As the timescale of development can be different between the two in general, the comparison at different time points is included.) The green line indicates the locations where the similarity indicator takes maximum for each developmental time step of individual A1 (horizontal axis). Note that the high similarity region is approximately 100–200 in either individual developmental time, which is neither the beginning nor the end of the developmental dynamics. (C) Projection of the similarity indicator on the green line in B against the developmental time of the first species (i.e., the horizontal axis in B).

time, and the vertical axis is the index indicating the cell location in a one-dimensional line. Different colors indicate different cellular states. The two individuals diverged from a common ancestral individual in the 800th generation. For precision, in this case, after selecting individual A1 in the primary simulation of 1000 generations, the ancestral individual of the 800th generation was chosen, and an additional simulation was performed over 200 generations (i.e., 1000 generations in total, equal to the generations in the primary simulation). Thus, owing to the secondary simulation, individual A2 was selected to represent different species diverged from a common ancestor.

By examining the ontogenetic change of spatial patterns with the differentiation of cell types, it was noted that there are some phases within which patterns change only slightly in ontogeny, and between them, drastic changes occur. Considering this case, there were three phases: early, intermediate, and late. By comparing the development of the two species (i.e., individuals with distinct GRNs), it may be difficult to judge at a glance in which phase the two are most similar to each other (Fig 2A). Thus, we quantitatively compared the developmental processes of the two individuals to examine the similarity in patterns between them throughout the course of development. The differences in gene expression between the species were computed across each cell of the same spatial location, and the inverse of the sum of the differences was used as an indicator of the similarity in gene expression patterns between the two individuals. This indicator is similar to those used in experimental studies [9–15,18,19], in which the average expression over the whole cell is adopted by transcriptome analysis whereas it is superior in that it includes spatially local information. (Note, however, that the same result was obtained even if we adopted the spatially averaged indicator as adopted in the experiments, see also S1 Fig).

Fig 2B displays the similarity between the gene expression states of individuals A1 and A2 at each developmental time. The indicator of similarity explained above was computed at each time point, as shown in the heat map. The horizontal axis represents the developmental time of individual A1, and the vertical axis represents that of individual A2. Fig 2C displays the time course of similarity, which indicates that the peak of the similarity is in the middle of the developmental dynamics (see also S2 Fig).

The similarity in gene expression patterns, as shown in Fig 2B and 2C, can be calculated for any two species obtained in this simulation. Additional examples are shown in the SF2. As long as the compared species were phylogenetically close, we observed the peak of the similarity between two species that diverged from a common ancestor, always in the middle of developmental time, which is consistent with the developmental hourglass [9–15,18,19].

Next, we examined the phylogenetic range within which the above similarity peak was observed by comparing the two species that diverged from earlier common ancestors. Fig 3 presents how the time course of the above similarity changes as the generations that the two species diverged earlier in the phylogenetic tree, where the similarity was computed between the original species and that departed at different branching points in the left phylogenetic tree (A, B, C). It was plotted against the developmental time of the original species, as shown in Fig 2C.

The peak of similarity in the intermediate developmental stage was most prominent in comparison with the species departed at branch A (i.e. separated at the most recent 200th generation), and observed for B, albeit a bit weaker (separated at the 500 generations before), whereas in C, that is, the comparison between species separated much earlier in the tree (at the 800th generation), the middle peak disappears (see S3 Fig, how the intermediate peak diminishes, as the point where the two species were increased per 100 generations). Hence, the similarity peak in the intermediate developmental time is preserved for comparison with the species that have diverged up to a certain depth range in the phylogenetic tree, but if they departed from a much farther ancestor, the peak disappeared.

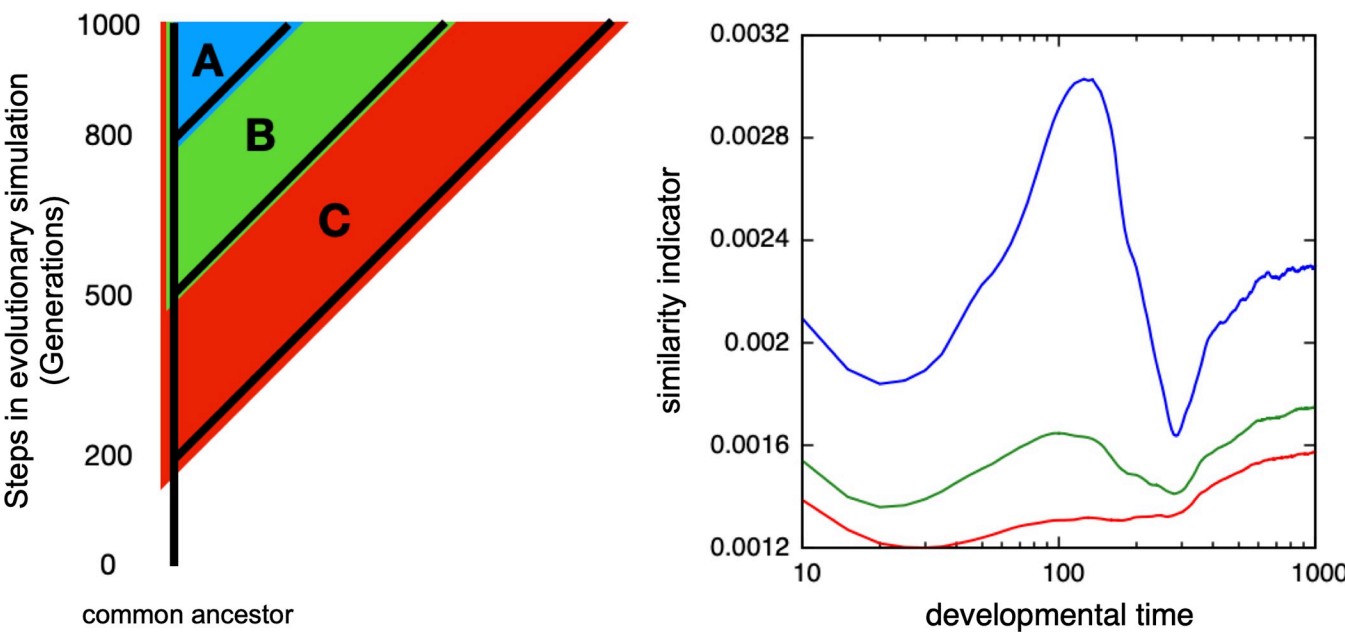

**Fig 3. The similarity between two species that departed from a common ancestor at given earlier generations in the phylogenetic tree.** The similarity is computed as in Fig 1B and 1C, the average over pairs of individuals. The horizontal axis represents the developmental time of the original species, and the vertical axis represents the similarity between the spatial patterns of gene expression. The line color denotes how far the two species are, in comparison, branched in the left phylogenetic tree, diverged from the 800th generation A(blue), 500th generation B(green), and 200th generation C(red). The horizontal axis was plotted on a logarithmic scale. At the intermediate developmental time, the similarity indicator takes a maximum when the pair branched at the later stage (in this example the peak exists up to ~500, whereas including other examples, the peaks exist for the pairs branched at the recent 100–300 generations). Typically, the closer the phylogenetic pair of individuals, the more prominent the peak in the similarity indicator.

Note that the common similarity peak is also observed even between the evolved individuals and their ancestors, for instance, as seen in S4 Fig, where we compared the species at the 1000th generation and its ancestor at the 500th generation. This will be because the mechanism leading to the similarity in the intermediate developmental stage, as will be discussed later, has been conserved after the acquisition and will be preserved after maximal fitness is achieved by evolution.

Next, we examined whether the present simulation results also contain other properties reported in the development of the hourglass-like conservation pattern.

In [11], it was observed that genes expressed in multiple stages during development (pleiotropic genes) were predominantly expressed at the bottleneck middle stage. Thus, we defined pleiotropic genes in our simulation and computed the ratio of the number of expressed pleiotropic genes to the total expressed genes. To compute it, we split developmental time into 12 stages and defined the "pleiotropic genes" as those that are expressed for more than half of the stages. Then, the fraction of expressed pleiotropic genes at each stage was plotted over the developmental stages. As shown in Fig 4, the fraction had a peak at the bottleneck time shown in the blue band. This increase in expressed pleiotropic genes is consistent with the experimental data (even though one might see arbitrariness in the above criterion for pleiotropic genes, the maximal fraction of expressed pleiotropic genes at the bottleneck time is insensitive to the details of the criterion, such as the threshold to be judged as expressed, and the way of partition of developmental time to stages).

The recent experimental report shows that the difference in gene expression patterns between clones is minimal at the developmental hourglass's bottleneck [35,36]. We then tried to confirm if this characteristic is available from the simulation results and calculated the

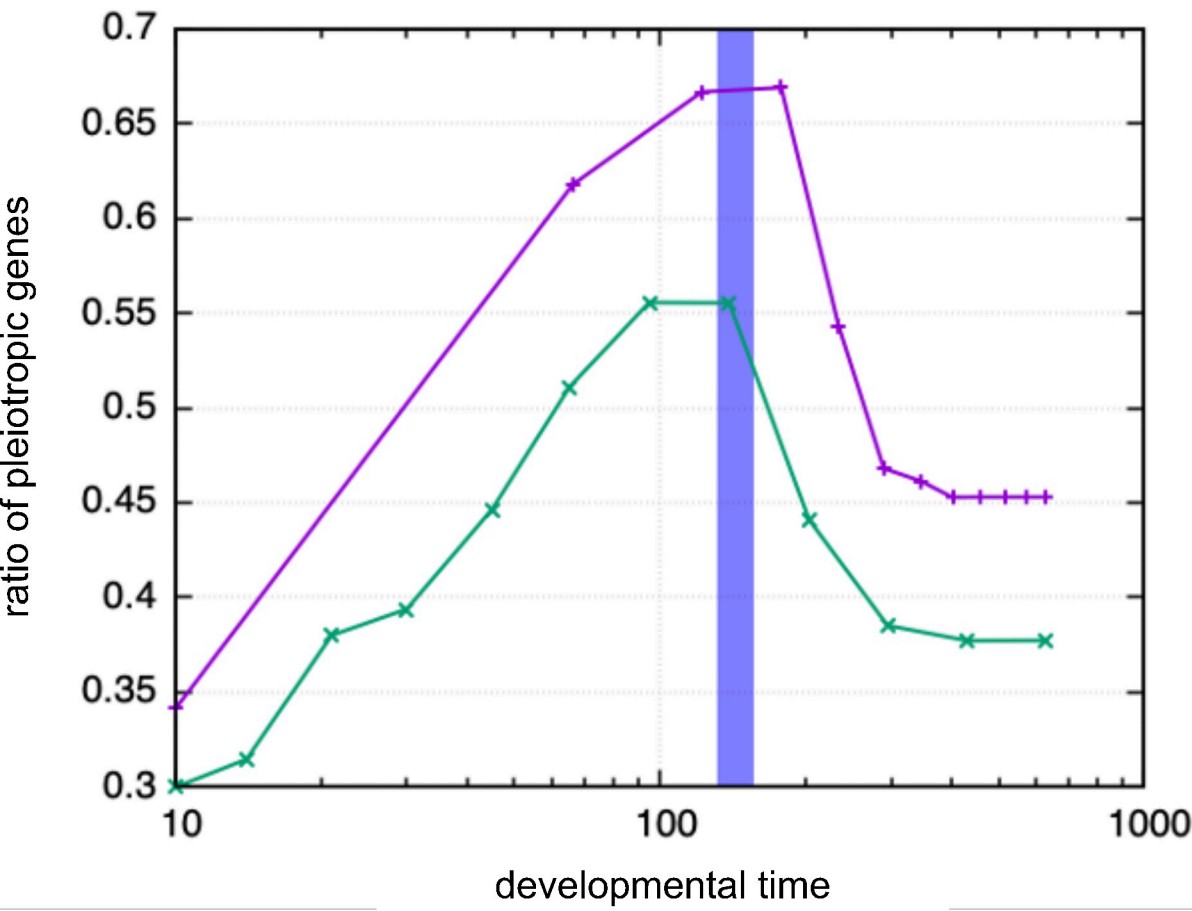

**Fig 4. The time course of the fraction of the expressed pleiotropic genes.** The pleiotropic genes are defined as those that are expressed for more than half of the developmental stages, whereas the stage was defined as splitting developmental time before reaching the stationary state into equal 12 spans, either by linear scale (the plot of purple (+) points) or by the logarithmic scale (the green (×) points). The fraction is obtained by averaging over 1000 individuals of different initial conditions, whereas the variance from the average was quite small. The area colored by blue indicates the typical range of developmental bottleneck (average ± standard deviation), which is given by comparison over other individuals diverted 200 generations earlier.

difference in the phenotypes (gene expression patterns) of clones of the same species, due to variation in initial conditions. This phenotypic variation of clones (i.e., the averaged gene expression difference between individuals from the same species), was computed in our model by taking different initial conditions (see Methods) is shown in Fig 5, against the developmental time in the logarithmic scale. The difference between individuals decreased up to a certain developmental time and then increased. The time that the variation took a minimum agrees with the time that the similarity in gene expression takes a maximum, i.e., the bottleneck of the developmental hourglass. (In other words, the average similarity between clone individuals takes maximal, statistically). After the variance takes minimal at the hourglass bottleneck time, it increases, as shown in Fig 5. In most cases, the increase is rather gradual towards the final stage of development, whereas a small peak is observed in some samples. As will be discussed below, due to a slight difference in the timing of the onset of the rise in the slow gene expression), which causes enhancement of the variance in some gene expression.

Note that the strength of the noise in the initial condition facilitates the emergence of a developmental hourglass. In this evolutionary simulation, the fitness is set up such that multiple cell types are generated in the final stage under the variation in the initial state. The

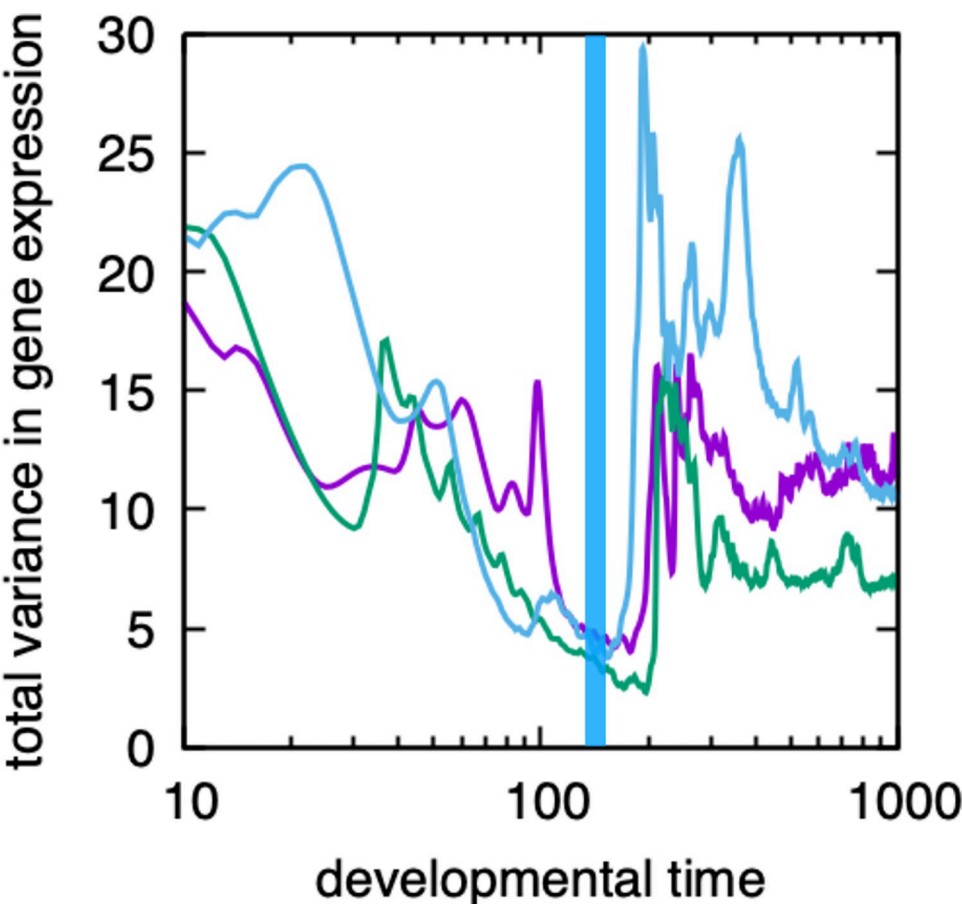

**Fig 5. The variance of gene expression patterns over 1000 clones, i.e., those generated from the same genotype but with different initial conditions.** The horizontal axis represents developmental time, and the vertical axis represents the total variance in gene expression over all cells, i.e., the summation of variance over cells, over genes, and over clones (see also Methods). As specific examples, the data of three different individuals are indicated by different colors. The blue stripe indicates the typical range of observed developmental bottlenecks (average ± standard deviation), as obtained by comparing individuals belonging to blue triangle A in the phylogenetic tree in Fig 3, which exhibits clear hourglass-like developmental correlations. Here, the initial conditions are chosen by adding Gaussian distribution noise with a variance $\sigma = 0.1$. (The details of the computation scheme are presented in the Methods section).

developmental process must reduce the initial variation to achieve robustness. Up to a certain developmental time, attenuation of the initial phenotypic variances by noise progresses. According to recent studies, this robustness to noise implies robustness to genetic changes [32]. Hence, while the variance within the same species is reduced during the developmental period, the variation in expression among genetically different species is also reduced (as long as they are not too far apart). This timing, in which the variance is minimized, agrees with the bottleneck in the hourglass (Fig 5). Furthermore, if we evolve the system by reducing the variance of the initial conditions (or noise), then the developmental hourglass is no longer observed (see S5 Fig).

In addition, we also computed the variance among individuals of the same network, due to continuously added noise throughout the developmental process. In this case, again, we observed that the variance due to the noise is minimal at around the bottleneck time.

Furthermore, to examine the universality of our result, we have carried out 21 runs for a set of evolution simulations of the present model, where different spatial patterns are observed

after reaching the fittest state. Among 18 of 21, we have observed the similarity in the intermediate stage, i.e., the developmental hourglass. (In fact, the exceptional 3/21 cases in which a clear peak was not observed contain oscillatory gene expression in the earlier developmental stage, which masks the peak). Also, it should be noted that in the sample runs showing the developmental hourglass, common peaks are observed for any pairs of species diverted from the common ancestor, as far as we have examined.

In summary, our simulation reproduces the hourglass properties reported in real evolutionary developmental biology (evo-devo) data thus far, despite the simplification adopted in the model.

## The developmental bottleneck and slow-gene control

Here, we note that the developmental similarity diagram (Fig 2) drastically changes in the middle developmental period, where the pattern itself changes in a discrete manner rather than continuously. Below, we explore how these drastic/discrete changes in development shape the developmental hourglasses.

Fig 6A illustrates examples of the changes in the expression of each gene in one individual over the entire space. This figure also confirms the stepwise change (Here, the gene expression remains at an intermediate level between on and off only if the input to it is close to the threshold for its expression. Switching between on and off is quite fast unless the gene's timescale parameter $\gamma_i$ is very small (see also Methods). Hence, for most genes, the expression changes are stepwise). Furthermore, there is a stage in which the expression of many genes exhibits discrete changes (switches). This stage coincides with the bottleneck time of the hourglass. Before the bottleneck time, gene expression exhibited little spatial heterogeneity as shown in Fig 6A. After the bottleneck, spatial subregions are generated in which specific genes are expressed and spatial heterogeneity is increased.

The time course of all gene expression levels for a given cell is shown in Fig 6B, which also demonstrates that the expression of various genes exhibits on/off switch changes around the bottleneck of the developmental hourglass stage. The expression levels of many initially expressed genes slowly decreased up to this bottleneck time, when fewer genes started to be expressed, depending on the spatial position. (The timing of the expression of specific gene sets in varies by individual, which causes a drastic increase in phenotypic variance in Fig 5).

The data also suggest the existence of a few genes whose expression levels change slower than those of the other genes. We computed the time scale of the expression changes of each gene (identical to the time for each gene that needs to change its expression from zero to fully expressed, or vice versa; see Methods for details) and plotted them in the order of their magnitude. As shown in Fig 6C, there are a few (four in the figure) genes whose expression levels change much more slowly than others. In the present paper, they are called 'slow genes'. Such slow genes have also been observed in previous studies, which are key to relating the developmental timetable with evolutionary change, as they tend to control the expression of other genes whose expression levels change at a much faster time scale [34,37]. Furthermore, Fig 6B suggests that the time scale of the slow gene is of the same order as the bottleneck time. This is reasonable because, with the time scale when expression levels of slow genes change, the expression of many other genes that are controlled by the slow ones are also switched between on and off.

In Fig 6D, the correlation between the timescale of these slowly changing genes and the average bottleneck time of the developmental hourglass is plotted. These variables are linearly correlated. To confirm the linear relationship, we also created mutants in which the timescale parameter $\gamma_i$ of slow genes was altered and examined if the bottleneck time changed

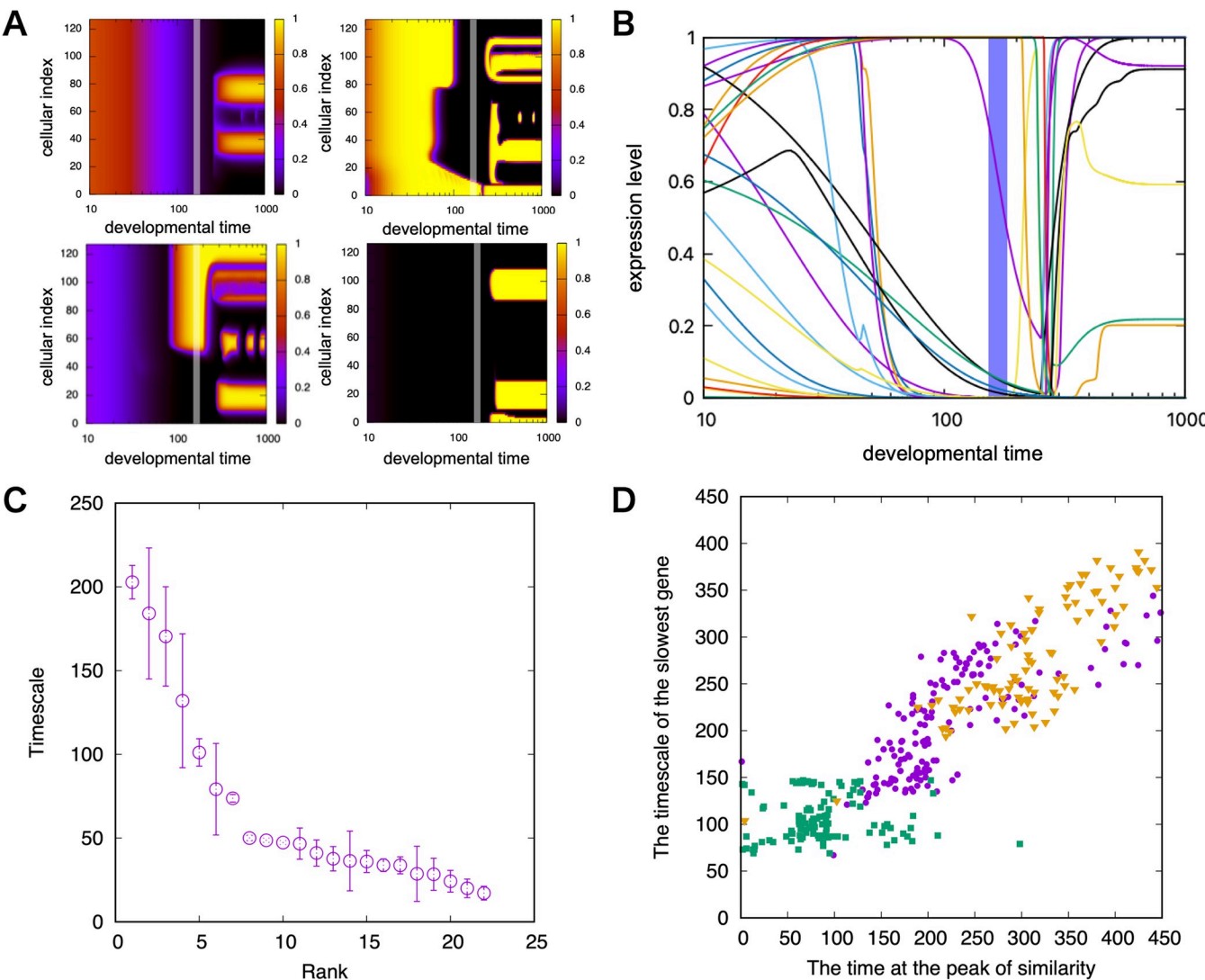

**Fig 6. Control of the developmental schedule by slow genes.** (A) Examples of spatial gene expression dynamics. The gray vertical line in each figure is the timescale of the genes whose expression changes are slowest (see C and Methods). The left upper corner is the expression dynamics of the slowest gene. (B) Time series of expression levels of all genes at cell index 32 in A. Different colors correspond to the expression levels of different genes. (C) Time scale of expression changes of each gene, which is plotted in order as a function of its rank in magnitude. Each circle denotes the average value of the timescale of genes over cells. Error bars are the standard deviation. (For the computation of the timescale of genes, see the Methods section). (D) Correlation between the timescale of the slowest gene and the timing of the bottleneck of the detected developmental hourglass. Magenta circles represent data from the individuals acquired through evolutionary simulation, whereas green squares and orange triangles represent data obtained by adopting a parametric change to the slowest gene in the original (blue dots). Green squares represent the cases in which the timescale of the slowest genes was changed to five times faster, and the orange triangles represent those changed with ⅕ times slower (See also [37] and S6 Fig for the heterochronic shift induced by manipulation of the slow gene).

accordingly. Even when such external change in the parameter was imposed, the common linear relationship between the timescale of slow genes and the bottleneck time was still observed (see Methods for the computation of the timescale of genes). This may suggest a possible causality from slow genes that control the developmental time schedule to the timing of the bottleneck of the developmental hourglass. Most individuals that shared a bottleneck in the developmental hourglass shared common slow genes. In contrast, the probability that two (distant) species that do not share the same developmental hourglass have common slow genes is much lower, and at a chance level, that is, the level for two randomly generated networks.

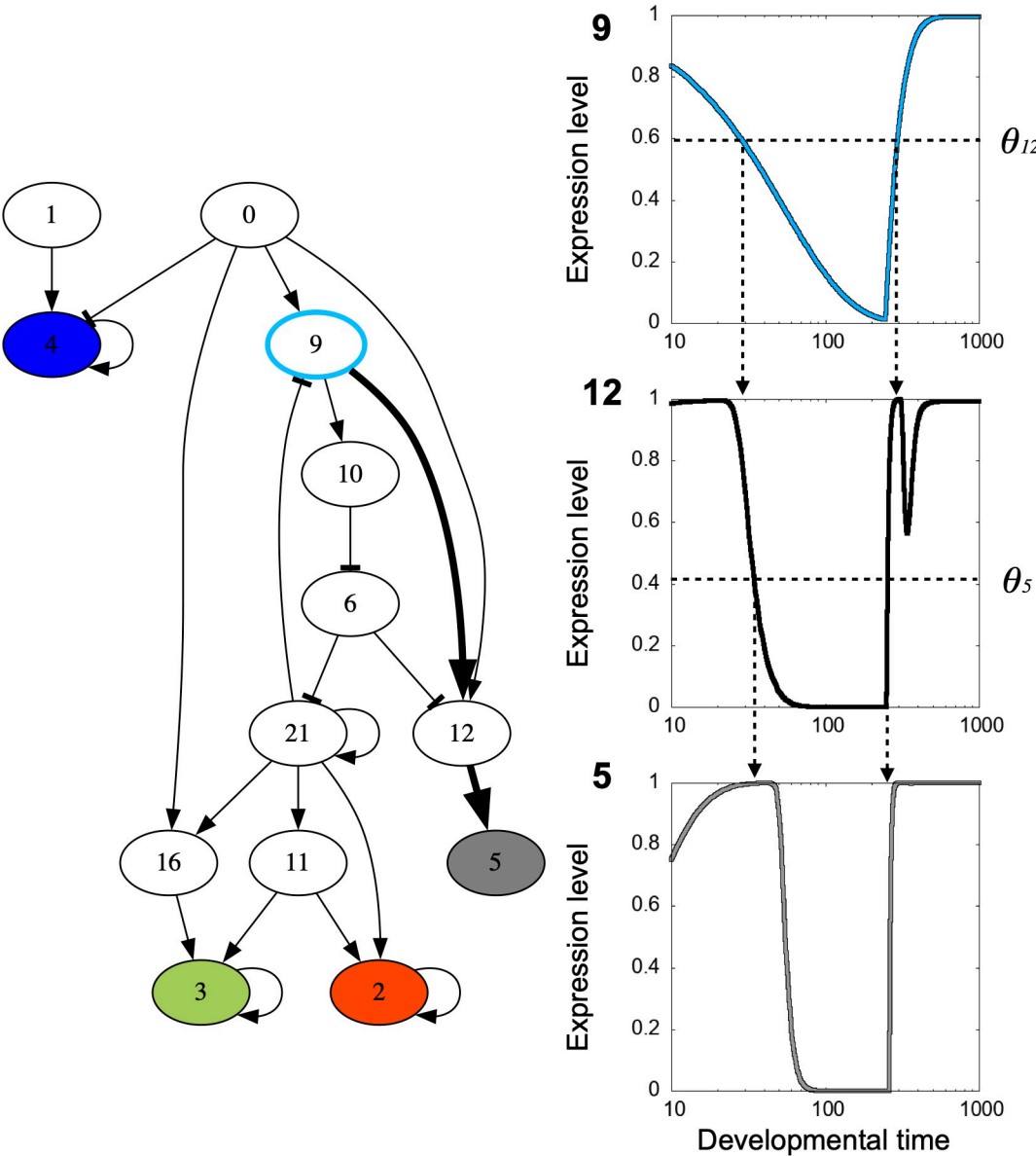

**Fig 7. An example of evolved gene network and expression control by the slow gene.** Left: The essential structure of the gene network of the individual of Fig 2A-1. Nodes filled with color are marker genes to determine fitness. Each color corresponds to the code in Fig 2A, i.e., red, green, blue, and brightness (which is represented as gray). The node with the light blue circumference is the slow gene. The path from the slow gene to the gray marker gene is shown as thick arrows. Note that paths from the slow gene to other marker genes exist. The procedure to extract essential gene network structure is explained in the Methods section. Right: Example of gene expression control of the slow gene. Expression dynamics of the genes in the thick arrowed path on the left, namely 9 (top), 12 (middle), and 5 (bottom), are shown for the cellular index 35. In each of these subfigures, the horizontal dashed line indicates the gene expression threshold of the arrowhead gene, and the vertical dashed arrows indicate the developmental time when the gene expression dynamics and the arrowhead gene expression threshold crossed, which basically corresponds to the expression change in the arrowhead gene.

To unveil if and how a slow gene controls the developmental time schedule, we explored how the gene works in the GRN. In Fig 7, we plotted the essential network structure extracted by the procedure described in the Method section. As shown, the slow gene controls the on/off pattern of target genes which lie downstream of the slow genes. Here, in contrast to most gene expressions that saturate with a faster time scale (~ 10 time steps), the expression of the slow

genes changes with a much slower time scale. Following its change, on/off expressions of other (including target) genes change quickly, as shown in the time series of Fig 7.

## Discussion

In summary, our evolutionary simulation exhibited the developmental processes consistent with the developmental hourglass. When we compared species diversified from the common ancestor, we observed (i) amongst phylogenetically close species, the similarity in the gene expression patterns is maximal at the intermediate developmental period. Even though neither the initial population nor the fitness condition favors the developmental hourglass, it was spontaneously shaped through evolution. (ii) at this bottleneck period, the number of pleiotropic genes was at the maximum, and (iii) the individual variances in gene expression levels over clones were at the minimum. These three features agree quite well with those experimentally reported as the developmental hourglass. We then uncovered that, at that stage, expressions of many genes are switched on or off, driven by a few genes whose expressions change slowly ("slow genes"). The timescale of the expression dynamics of these slow genes agrees with that of the bottleneck period.

To gain robustness in the developed pattern, unavoidable variations that existed in the initial stage and during later developmental dynamics should be reduced through the developmental course. In our model, such reduction progresses up to the bottleneck stage, where the variation among individuals is minimized. In fact, when there is a sufficient level of variance in the initial condition, as adopted in our simulation, the robustness to the variation evolves, and in this case, the developmental hourglass is observed, whereas if the initial variance was set to be too small, the developmental hourglass was not observed as in S5 Fig.

In recent theoretical and experimental studies, the correlation between phenotypic variance by noise and that by genetic changes has been observed [32,38,39]. There, robustness to noise fosters the robustness to mutation, i.e., genetic assimilation of robustness was observed. By extending the argument of such a correlation to the developmental time course, one may expect that the reduction of the initial variance by noise leads to a reduction in variance across genetically altered individuals. This explains the coincidence between the time of minimal variance over clones and the developmental bottleneck over genetically different species, which was recently suggested [35,36]. The underlying mechanism for obtaining the robustness to noise in the initial state needs to be further elucidated, while the idea of the slow gene control found and discussed in this paper may provide a clue to uncover it.

At the bottleneck stage, the expression of slow genes sets in, and most genes change their expression levels. Following the expression of slow genes, other genes are diversified by species, leading to a developmental hourglass. The later developmental processes are contingent on the species. However, the mechanism driven by slow genes to absorb variations in the initial variation was preserved through evolution. Thus, the developmental hourglass is expected to be observed for species with the shared control mechanism mediated by slow genes.

In summary, the developmental hourglass obtained in our simulation results was caused by an attenuation of the variation in the initial state, accompanied by the expression of a few slow genes that control the developmental process. If these slow genes are conserved among species that diverged from the common ancestor, they share the common developmental bottleneck time determined by the slow genes.

If the two species diverted far earlier, the similarity peak would be diminished. This decrease might look gradual at first glance, but indeed, at a certain point, the peak disappears distinctively. This point roughly agrees with the point at which the commonality of slow genes is lost as the two species branched earlier. Here, the attenuation of initial variation should always be present, but the species separated earlier would use different gene sets for the slow

process, and then the gene expression patterns of the two species would no longer show similarity in the intermediate developmental stage, and the similarity peak would no longer exist. In this case, the similarity can be maximal either at the very beginning or at the last developmental time (Fig 3). This result may correspond to the cross-phylum comparison of gene expression patterns, as reported previously [40] (whereas we note the criticism to it [41] as discussed below). It will be important to examine closely within which range the common hourglass behavior is observed.

The slow genes regulate the expression of many other genes, as was shown previously [34]; thus, with this timescale of slow genes, many genes change their expression levels, including pleiotropic genes. This leads to an increase in the fraction of expressed pleiotropic genes in the developmental bottleneck period, which is consistent with the present simulation and experimental data [11]. The phylogenetically conserved time scheduling by slow genes can explain the range of species in which the hourglass was observed, which might correspond to the range within the same phylum. In the present simple model, morphogenesis is determined only by the expression of genes; thus, the control of development is regulated solely by gene expression, whereas current experimental studies revealed that mechanical interactions and subsequent cell motions are important in the development of real multicellular organisms [42]. When such processes are included, the commonality of the slow control mechanisms (not restricted to gene expression dynamics) may explain the hourglass.

The idea of the developmental hourglass stems from the discovery of the conservation of specific gene expression patterns in animals [4]. Conceptually, the bottleneck of the developmental hourglass is thought to correspond to the stage at which morphological characters that define the phylum of the species arise, the so-called phylotypic stage [5,6].

After the finding of the developmental similarity was made, the reverse question was posed, i.e., whether a group of animals that share the developmental hourglass agrees with a common phylum [43]. This question remains unanswered, which will be important in uncovering common developmental mechanisms that provide a basis for the developmental hourglass. Our results may suggest that species that shape the developmental hourglass could be defined as those that share a common slow control mechanism to reduce noise in the early developmental period. Exploring such common slow mechanisms would be important, even though the species that form a developmental hourglass may not correspond to a phylum.

The study's primary limitation is that the model employed in this simulation is extremely simplified in terms of the number of genes and spatial degrees of freedom compared to real organisms. In particular, only gene regulations and diffusive cellular interactions are considered. Other layers of development, such as the collective movement of cells under mechanical forces, reported as important in tissue morphogenesis and responsible for making shape, are not considered [42]. However, the synergy between gene expression dynamics and cell-to-cell diffusion is common in the development of real organisms [44] and when represented mathematically, such mechanical processes can also be represented by a combination of gene expression dynamics and diffusive interactions among cells [45]. The basic results obtained in this study will be valid under the inclusion of such processes.

## Methods

### Model

The model in this study considered the evolution of a finite-sized population of 100 individuals. An individual is defined as 128 cells allied in a one-dimensional space without cell proliferation during gene expression dynamics. In each generation, individuals exhibit specific gene expression patterns according to gene expression dynamics with cell-cell interactions provided

**Table 1. List of the parameters adopted in the simulation.**

| | Description | Change in each generation by mutations |
|---|---|---|
| $J_{i,j}$ | Gene network structure matrix | Randomly chosen 4 $(i, j)$ pairs are subject to the mutation with the probability 1/8 (for the change to 1 or -1) and 3/4 (for the change to 0). |
| $\gamma_i$ | The timescale of the expression dynamics of gene $i$ | Gaussian noise with which the average is 0 and the standard deviation is 0.01 is added to the value. |
| $\theta_i$ | The expression threshold for gene $i$ | |
| $D_i$ | The diffusion constant of gene $i$ product | |
| $x_i^{init}$ | Initial value of gene $i$ expression | |

| | description | value |
|---|---|---|
| $N$ | The number of genes in each gene network | 24 |
| $L$ | The number of cells in each individual | 128 |
| $\beta$ | The nonlinearity of the reaction near the threshold | 40 |
| $P$ | The number of individuals in each population | 100 |
| $t_{term}$ | The number of timesteps for development | 1000 |
| $G_{term}$ | The number of generations for the evolutionary simulation | 1000 |
| $\sigma$ | The strength of the epigenetic perturbation on initial condition | 0.1* |

*This value is changed in SF4

by their own gene regulation networks. For the gene expression patterns of each individual, fitness is computed, and those with higher fitness can produce multiple offspring, and those with lower fitness cannot. Therefore, 100 descendants are generated in total. Evolutionary simulations were conducted by iterating the scheme. This section provides a detailed description of our proposed model, including the set of parameters adopted in the simulations (Table 1), for the reproduction of the results.

## Gene regulation networks

In this study, the total number of genes in gene networks was fixed at $N = 24$, and gene regulation networks were described as an $N \times N$ matrix $J_{i,j}$, where the interaction between gene $j$ and gene $i$ is given by its $(i, j)$ element. Interaction was represented as a discrete value of 1 (activation), -1 (inhibition), or 0 (no interaction between the two genes).

## Gene expression dynamics

A cell's state is represented by the expression levels of $N$ genes, that is, the number of expressed proteins. Here, let $x_i(l, t)$ represents the protein expression level of the $i$-th gene in the $l$-th cell at time $t$. Cells are aligned in a 1-dimensional space. Since this model does not employ cellular proliferation, cellular index $r$ assigned to each cell is fixed throughout the simulation, which takes an integer value ranging from 1 to the total number of cells $L$ ($= 128$). Proteins expressed from each gene either activate, inhibit, or do not influence the expression of other genes or their own expression. This protein level is als termed as gene expression level. We assumed that the change in the $i$-th protein expression level is given by the following *discrete-space* reaction-diffusion equation:

$$\frac{\Delta x_i(l, t)}{\Delta t} = \gamma_i(F(i, \boldsymbol{x}(l, t)) - x_i(l, t)) + D_i(x_i(l + 1, t) - 2x_i(l, t) + x_i(l - 1, t)) \tag{1}$$

The first term of Eq (1), $\gamma_i(F(i, \boldsymbol{x}(l, t)) - x_i(l, t))$ denotes the change in protein expression by reactions within a cell. This term consists of two parts, i.e., $F(i, \boldsymbol{x}(l, t))$ and $-x_i(l, t)$. The former represents the protein synthesis under regulations of other proteins and the latter represents the degradation of the protein. The expression of protein $i$ is regulated by other proteins $j$. As are often adopted in simplified gene regulation dynamics models, $F(i, \boldsymbol{x}(l, t))$ is a result of activation or suppression from other proteins. If the input from others is larger than given threshold, it is tended to be expressed, and otherwise not expressed. The specific form we adopted is given by the following equation.

$$F(i, \boldsymbol{x}(l, t)) = f\left(\sum_j J_{i,j} x_j(l, t) - \theta_i\right) = \frac{1}{1 + e^{-\beta(\sum_j J_{i,j} x_j(l,t) - \theta_i)}} \qquad (2)$$

where $x(l, t)$ denotes the vector of the expression of each gene $x_i(l, t)$ ($i = 1,2,..N$).

In the above equation, $J_{i,j}$ changes by generation $k$ through evolution, and thus it has the index $k$, but putting the index $k$ in $J_{i,j}$ makes the expression too complicated, so it is omitted here. The function $f(z) = 1/(1 + e^{-\beta z})$ is similar to a step function, where the function approaches 1 as $z$ is increased to the positive side and approaches 0 as $z$ decreases to the negative side; in other words, in Eq (2), if the total sum of gene interactions $\sum_j J_{i,j} x_j(l, t)$ is sufficiently larger than the expression threshold $\theta_i$, then $F(i, \boldsymbol{x}(l, t)) \sim 1$, which indicates that the gene is going to be fully expressed; if it is smaller than $\theta_i$, then $F(i, x(l, t)) < 1$, which indicates that the gene expression is suppressed. We choose $\beta$, the sensitivity of the function $f(z)$ at $z = 0$, as 40. In general, $\beta$ is proportional to the Hill coefficient.

$\gamma_i$ determines the rate of the degradation of the $i$-th protein, whereas the value of $x_i(l, t)$ was scaled such that the maximum level was equal to 1 [27,46,47].

The second term in Eq (1), $D_i(x_i(l + 1, t) - 2x_i(l, t) + x_i(l - 1, t))$ is a discrete-space version of the diffusion, which denotes cell-to-cell interaction via diffusion where $D_i$ is the diffusion constant of the product of the gene $i$, that determines the rate.

The numerical timestep interval $\Delta t$ is set to 0.1.

## Generation and density of gene regulation networks

In this study, gene regulation networks were initiated using a semi-random generation process. For each element of the network matrix, interaction values of -1, 0, and 1 are assigned probabilities of 1/16, 7/8, and 1/16, respectively. The generated networks have network density, that is, the number of total connections divided by the total number of genes ~ 2, which is comparable to the real data [48–50].

## Initial/boundary conditions

Morphogen gradients, which were introduced as the expression of genes #0 and #1, were supplied externally. Fixed linear morphogens were induced from both sides for cellular use, such that $x_0(r, t) = x_0(t) = (L - l)/(L - 1)$ and $x_1(l, t) = l - 1/(L - 1)$, where $L$ is the total number of cells that is set to 128 in the simulations and $l$ is the cellular index from 1 to $L$. These gradients are not only present in the initial state but are invariant in time. We also evaluated a case involving a gradient with an exponential dependence in space as $\propto e^{-\frac{l}{\xi}}$, but this condition did not alter the conclusions presented in this study.

For genes other than the input genes #0 and #1 (i.e. genes #2, #3,. . .,#N), the initial expression levels of each gene (at $t = 0$) were given spatially uniform. The initial value is chosen as

$$x_i(l, 0) = x_i^{init} + \delta_i$$

where $x_i^{init}$ is a value inherited from the ancestor slightly altered by mutations (as described later), whereas $\delta_i$ is the deviation due to the epigenetic perturbation, given by a random value from the Gaussian distribution with a mean 0 and the variance $\sigma$, independently generated for each gene, but taken uniformly over all cells. This noise term is due to environmental, behavioral, and other non-genetic effects, so that the initial condition is deviated from the inherited value $x_i^{init}$. Except for the SF4, $\sigma$ is set to 0.1.

Discrete Neumann boundary conditions were adopted at both ends for this study, that is $x_i(1) = x_i(2)$ and $x_i(M-1) = x_i(M)$ for $i = 2,3,4,....N$.

## Fitness function

We defined cell types according to the ON/OFF states of the designated four genes (genes #2 to #5 in the model). Specifically, if one gene's expression is over 0.99 (with the value range [0,1]), the gene is ON, whereas if it is less than 0.01, the gene is OFF. The number of existing cell types, $N_t$, was computed by sampling over cells. The fitness function is set to become higher as more cell types coexist.

The model can maximally generate $2^4 = 16$ cell types such that the fitness of individual $k$ is defined as $f_k = N_t$. Hence, the fitness is maximal if 16 different cellular states exist in cells aligned in a one-dimensional space.

With this setup, the expression level of each cell can settle to an intermediate value, between 0 and 1, and thus the expression levels of the target four genes can take intermediate values, between ON and OFF. However, since gene expression sensitivity $\beta$ is set to 40, which corresponds to a quite high Hill coefficient, $x_i(l, t)$ of the four target genes takes either ~1 (ON) or ~0 OFF for the final, stationary state of the developmental dynamics. (The exceptions are the case when the input to the corresponding gene is close to the expression threshold $\theta_i$ ($\sim \frac{1}{\beta}$), and such case is not favored under the evolution of the present fitness, because the intermediate expression levels of the target genes just reduce the number of possible cell types. In fact, after evolutionary generations by the selection, such exceptions are hardly observed, except few cases with oscillatory gene expression localized at some cells)

## Selection

Whether an individual in a population is selected for the next generation is determined probabilistically. The probability that one individual would be selected is given by its relative fitness to others, i.e., $e^{f_i} / \sum_j e^{f_j}$, for all $j$, where $i$ and $j$ are individual indexes. We adopt this exponential form to amplify the selection by fitness, i.e., to amplify the advantage of having one more cell type than other individuals (By simply adopting $f_i / \Sigma_j f_j$ as the probability, the advantage of having one more cell type over other individuals was too small and the individual with one more cell type was hardly selected.

In fact, such exponential form is often adopted in genetic algorithm or numerical evolution.

## Mutation

During the replication process, the gene network structure and parameters in the reaction-diffusion equation were slightly changed by the random mutation. Here is the list of evolvable parameters and their degrees of change during one replication.

## Population cloning

Each evolutionary simulation was executed for up to 1000 generations, and we sampled cases in which the fitness of the fittest individual at the 1000th generation was above 15 (the

maximum was 16). For these fitted individuals, we further cloned the ancestral populations. We selected ancestral populations from each 100th generation and evolved them up to 1000th generation. The parameters and mutational setups for the evolutionary simulations were identical to those of the original simulation.

Individuals obtained by these evolutionary simulations of the cloned population represent offspring that diverged from the same pedigree with different branching points.

## Calculation of the similarity indicator of gene expression patterns

The similarity of gene expression patterns between two individuals was calculated using two indicators: Pearson's correlation coefficient and the inverse of the total gene expression difference. The definitions of the two indicators are as follows:

1. the inverse of the total gene expression difference

$$\Delta(t_1, t_2) = \frac{1}{\sum_i \sum_r |x_i^1(l, t_1) - x_i^2(l, t_2)|}$$

where $x_i^k(r, t)$ indicates gene $i$'s expression level at cell $l$, time $t$, of the individual $k$.

2. Pearson's correlation coefficient
For each developmental time $t_1$, $t_2$, the correlation coefficient $\rho(t_1, t_2)$ is given as

$$\rho(t_1, t_2) = \frac{cov(\boldsymbol{X}^1(t_1), \boldsymbol{X}^2(t_2))}{\sigma_{X^1(t_1)} \sigma_{X^2(t_2)}}$$

where $\boldsymbol{X}^k(t)$ is the vector in which elements $X_i^k(t)$ are the summation of gene $i$'s expression over space in individual $k$ at a given time $t$, that is, $X_i^k(t) = \sum_r x_i^k(r, t)$, $\sigma_{X^k(t)}$ is the standard deviation of $X^k$ at time $t$, and $cov$ denotes the covariance.

## Identifying pleiotropic genes

Following the experimental report in ref [11], pleiotropic genes in this paper refer to the genes expressed in multiple developmental stages. The numerical procedure to identifying them is as follows:

First, we defined developmental time points in which gene expressions are compared. In ref [11], these time points are 16 embryonic stages determined by staging table of development. We defined stage time points as arithmetically or logarithmically even intervals of 16 time points through [10,1000].

For each developmental stage time point $s_j$, the average expression in space was calculated for each gene.

$$\overline{x_i(s_j)} = \frac{1}{L} \sum_{l=0}^{L-1} x_i(l, s_j)$$

where $\overline{x_i(s_j)}$ is the average expression of gene $i$ over space at stage $s_j$.

Then we set a threshold to determine if a gene is expressed or not, which is supposed to corresponds to the detection limit in experiments so that we set it small as 0.1.

If $\overline{x_i(s_j)}$ is larger than the above threshold 0.1, for more than the half of the number of total stages (i.e., 8 = 16/2), the gene is determined as a pleiotropic gene.

The ratio of the pleiotropic genes plotted in Fig 4 was defined for each stage time point as the total number of expressed pleiotropic genes divided by the total number of expressed

genes. Even if the number of stage time points or the threshold value was changed, the result presented does not change, as long as the change was not too large.

## Calculation of variance in the pattern dynamics over clones

For a given individual, i.e., a set of GRN and parameters, $N$ genetic clones are made while adding Gaussian noise to the original initial gene expression state of the development. The average and the standard deviation of the noise are 0 and 0.01, respectively. The developmental dynamics of those clones are calculated according to the same equation for gene expression dynamics. Then, the variance in the pattern dynamics over clones, which indicates how the initial difference propagates during development is calculated as

$$var(t) = \frac{1}{N} \sum_k \sum_l \sum_i \left(x_i^k(l,t) - \overline{x_i(l,t)}\right)^2$$

Where $Var(t)$ indicates the developmental variance over clones at time $t$, $x_i^k(l,t)$ is the gene $i$'s expression in an individual $k$, at cell $l$ and time $t$. $\overline{x_i(l,t)}$ is the average of $x_i(r,t)$ over clones, i.e.,

$$\overline{x_i(l,t)} = \frac{1}{N} \sum_k x_i^k(l,t)$$

## Estimation of the timescale of slow gene

To estimate the timescale of genes, first, we defined a unit of time step. The unit is set to be small enough so that the change within the unit is small. In this study, we adopted the unit as the interval for the increment of time during the numerical simulation of gene expression dynamics. Similar calculations could be performed in actual experimental systems if the interval was chosen sufficiently small compared to the time needed for gene expression to saturate.

By using this unit, we can estimate the partial derivative of gene expressions as $\frac{\partial x_i(r,t)}{\partial t}\big|_{t=t_k} \sim \frac{x_i(r,t_k)-x_i(r,t_{k-1})}{\Delta t}$, where $x_i(r,t)$ is the gene expression level of gene $i$ at location $r$, time $t$. $\Delta t$ is the timestep and $t_k$ indicates the time after the $k$ timesteps, i.e., $t_k = k\Delta t$ ($k = 1,2,\ldots$).

The timescale of genes can be estimated by counting timesteps according to the following procedure.

Count up $k$ by 1 if the following two conditions are not violated.

1. The sign of the derivative at $t = t_k$ is the same as that of the time at $t_{k-1}$. i.e.,
   $$\frac{x_i(r,t_k)-x_i(r,t_{k-1})}{\Delta t} \frac{x_i(r,t_{k+1})-x_i(r,t_k)}{\Delta t} > 0$$

2. The absolute value of the derivative at time $t_k$ is larger than a threshold; in this study we adopted the threshold as 0.001.

The condition 1 is for discarding oscillatory behavior and the condition 2 is for discarding time steps after saturation.

## Extraction of essential gene network structure

In our model, the gene network is identical for each individual and over cells. However, due to the heterogeneity of input morphogen gradients, different genes are activated, and different parts of the whole network are activated for different cells (leading to effectively different gene expressions dynamics by cells). Therefore, although the whole gene network contains all the information needed to regulate the dynamics, it is unsuitable for analyzing the gene regulations responsible for a cell. The whole network has too many 'decoy' edges that may work in

other cells, but do nothing in the cell under consideration. We extracted an essential network from the whole network using the following method to identify which regulations are essential in a given cell.

1. Trace the dynamics of the inputs to the marker genes, i.e., $\sum_j J_{i,j} x_j(t) - \theta_i$ $(for\ i = 2, 3, 4, 5)$ in that cell and detect the periods during at least one of the input is in the range of $[-2/\beta: 2/\beta]$, which is the dynamic range for the gene expression dynamics and correspond to the timing of gene expression changes of marker (see Eqs (1) and (2)).

2. Trace the expression dynamics of genes. If a gene is not expressed at all throughout the periods, the regulation edges from and to the gene are not responsible for the expression change in marker genes. Hence, edges from/to such genes are eliminated from the whole network. Also, if the expression level of gene stays constant throughout the periods, regulation edges from that gene just contribute to increase/decrease of the expression threshold of the arrowhead gene. Thus, edges from such genes are eliminated from the whole network.

3. For each remaining edge, check whether input from the root gene's expression contributes to the expression of the arrowhead gene. If the contribution, defined and computed below, is larger than a certain threshold value, then the edge is assumed to be responsible for the change. Otherwise, the edge is eliminated from the whole network.
Here the contribution is estimated as follows. For a given cell, let $\Delta x_{k,l}$ be the expression change of gene $k$ during the $l$-th period. Then the contribution of the edge $j{\rightarrow}i$ for the gene expression change in the $l$-th period is defined as

$$contribution:\ \frac{J_{i,j}\Delta x_{k,l}}{\sum_k J_{i,k}\Delta x_{k,l}}$$

If the contribution is smaller than a threshold, which was set to 0.01, delete the edge for input from the edge does not contribute to the total change in the input.

5. Last, delete all the edges on genes that have a route neither from morphogens nor to the marker genes.

## Supporting information

**S1 Fig. Developmental similarity of the examples in the main Fig 1 measured by different similarity indicators.** The similarity between the developmental dynamics of the two individuals shown in Fig 1A is represented as a heat map. The axes are the same as in Fig 1B, and the similarity is measured by Pearson's coefficient on the left and Spearman's rank coefficient on the right.
(TIF)

**S2 Fig. Additional examples that show developmental-hourglass-like similarity profile.** Four additional examples that show developmental hourglass are represented with developmental similarity. (Left) Two examples of gene expression pattern dynamics of evolved GRNs. The horizontal axis represents the developmental time, and the vertical axis represents the cellular index, i.e., space. (Right) The dependence of similarity is plotted against the developmental time, for species with different phylogenetic distance. The horizontal axis represents the developmental time, and the vertical axis represents the average of the similarity between the spatial patterns of gene expressions, in the same way as in Fig 2. The similarity is computed as

in Fig 1C and averaged over pairs of individuals. The line colour denotes the phylogenetic distance of compared species departed from 900, 800,. . ., 100, 0th generations. The similarity calculated among the species that diverged every 100 generations, from the 900th to the 0th generation is plotted.
(TIF)

**S3 Fig. The time course of the similarity indicator and its dependence on the phylogenetic distance.** Similarities among the species that diverged from a common ancestor are plotted, using the data from the same simulation run as in Fig 2, but plotted with phylogenic distance per 100 generations. Axes are the same as in Fig 2(left). Each curve represents the similarity among species that have different length of shared generations in the evolutionary path. The top blue curve represents the similarity of species that share 900 generations out of total 1000 generations, and with each 100 generations decrease of the length of shared generations, results are plotted as different colours, from blue to green (see also Figs 1 and 2 for schematic representation).
(TIF)

**S4 Fig. Similarity between an evolved species and its ancestors.** Similarities are plotted for the individual of Fig 2A1 with each 100-generation distant ancestor from the 900th to 0th generation. The horizontal axis is the developmental time of the evolved individual, and the vertical axis is the similarity indicator. The similarity is plotted in the same manner as in Fig 2C. The intermediate similarity peak is conserved from the 700th generation ancestor (i.e., 300 generations distant ancestor), which roughly corresponds with the slow gene control acquisition generation.
(TIF)

**S5 Fig. Examples of the similarity profiles with smaller noise levels for the initial condition.** The relationship between phylogenetic distance and the value of the similarity indicator is represented as the same as Fig 2, SF2, and SF3 but with different noise levels on the initial condition. (A) An example of the case with the noise level on the initial condition $\sigma = 0.01$, which is 0.1 in the other figures. (B) An example of the case with the noise level on the initial condition $\sigma = 0.001$.
(TIF)

**S6 Fig. Slow gene control and heterochrony.** Subfigures A)–C) display pattern formation dynamics of the same gene network except for the change in the time-scale parameter of the slowest gene. A: Pattern formation dynamics originally obtained from evolutionary simulation. The value of the timescale parameter of the slowest gene is c$5.17 \times 10^{-2}$. B: Pattern formation dynamics with a larger timescale of the slowest gene; $10.3 \times 10^{-2}$, 2 times larger than that of A. C: Pattern formation dynamics with a smaller timescale of the slowest gene; $2.58 \times 10^{-2}$, 0.5 times smaller than that of A.
(TIF)

## Acknowledgments

The authors would thank Masahiro Uesaka and Naoki Irie for their stimulating discussions and for critical reading of the manuscript.

## Author Contributions

**Conceptualization:** Takahiro Kohsokabe, Kunihiko Kaneko.

**Data curation:** Takahiro Kohsokabe.

**Formal analysis:** Takahiro Kohsokabe.

**Funding acquisition:** Shigeru Kuratanai, Kunihiko Kaneko.

**Investigation:** Takahiro Kohsokabe, Kunihiko Kaneko.

**Methodology:** Takahiro Kohsokabe, Shigeru Kuratanai, Kunihiko Kaneko.

**Project administration:** Shigeru Kuratanai, Kunihiko Kaneko.

**Software:** Takahiro Kohsokabe.

**Supervision:** Kunihiko Kaneko.

**Validation:** Takahiro Kohsokabe.

**Visualization:** Takahiro Kohsokabe.

**Writing – original draft:** Takahiro Kohsokabe, Kunihiko Kaneko.

**Writing – review & editing:** Takahiro Kohsokabe, Shigeru Kuratanai, Kunihiko Kaneko.

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
