## [Decision Letter · Decision Letter 0]

13 Aug 2023

Dear Prof. Kaneko,

Thank you very much for submitting your manuscript "Developmental hourglass: Confirmation by numerical evolution and elucidation by dynamical-systems theory" for consideration at PLOS Computational Biology.

As with all papers reviewed by the journal, your manuscript was reviewed by members of the editorial board and by several independent reviewers. In light of the reviews (below this email), we would like to invite the resubmission of a significantly-revised version that takes into account the reviewers' comments.

We cannot make any decision about publication until we have seen the revised manuscript and your response to the reviewers' comments. Your revised manuscript is also likely to be sent to reviewers for further evaluation.

Sincerely,

Philip K Maini

Academic Editor

PLOS Computational Biology

James O'Dwyer

Section Editor

PLOS Computational Biology

Reviewer's Responses to Questions

**Comments to the Authors:**

Reviewer #1: Kohsokabe et al. present an unusual approach to provide some mechanistic insight into the hourglass model of morphological evolution through simulation. They present an admittedly simple simulation framework that implements simple gene regulatory networks in 1D organisms with cell intrinsic and cell signalling effects. They simulate their ontogeny and evolve them over 1000 generations. Species are periodically split off from the simulated organisms to provide comparative data. Noisy initial conditions and mutations in GRN matrices at species transitions are implemented. Cell type diversity (under simplified, but strict definition) is used for selection of individuals between generation. They then develop a framework to compare spatial gene expression patterns that evolve in this simulated system across developmental time. This provides the ingredients for testing the predictions of the hourglass model. They find that indeed, in this system, mid stages of the developmental trajectory are more conserved between species compared to earlier and later stages, mid stages are enriched for pleiotropic genes (under the definition of the toy model) and show less divergence within species evolved from the same starting conditions (with noise). All this seems to mimic the general predictions of the hourglass model. Detailed analysis of the evolving GRNs indicate that the pattern could be driven by relatively few genes that change their expression slower relative to other genes. This is interpreted in the context of the theoretical work previously published by the authors on the subject.

The assumptions of the simulation seem reasonable. It is fairly difficult to understand the Figure 2 visualisation of the two species and how they arose, however, once the distinction between ontogeny (developmental time, plotted) and phylogeny (the number of generations that led to this individual and potential split along the path) sinks in, the paper is relatively easy to follow. The hourglass pattern is much stronger for more related species and all but disappears when the bifurcation appears early in the simulation. This is not necessarily the prediction of the hourglass model. The authors should discuss this. Also, given the obvious distance between this simple toy model and real evolving organisms, the authors should think about other possible, perhaps more trivial reasons for why the mid-simulation pattern convergence occurs. The gradual decrease in gene expression noise in the system is intriguing. The sudden rise of complexity in the system after that would require more thorough investigation. The authors make the connection with slow genes. However, this idea is not sufficiently developed. Would there be a way to investigate why does this dynamics coupling occur in this system? And if yes, could there be drawn some analogy with GRNs in real biological systems?

Overall, this is an unorthodox view of the hourglass model. Understanding the toy model and looking at its behaviour could be of interest to scientists who are in general struggling with the meaning, significance and mechanisms of this peculiar embryological observation.

Minor comments: The paper contains several typos. Figure 1: developmetnal time 2x, Figure 4 is not referred to at all, the relevant call out goes to Figure 3 again. Fonts in Figures are very diversified, different fonts and sizes.

Reviewer #2: Phenotypic evolution has been driven not only external factors such as environmental conditions, but also by internal ones, i.e., developmental processes. One of the key interests in evolutionary developmental biology lies in deciphering the evolutionary constraints and principles imposed by embryogenesis. The paper under review tackles this intricate issue by utilizing evolutionary simulations, a noteworthy methodological choice that distinguishes this study within the evo-devo field. This approach allows the authors to gain a new valuable insight into a problem that has been traditionally addressed by speculation based on observations of extant species and fossil records.

However, the paper presents several significant concerns. The most prominent point is the ambiguity in the presentation of the methods, data interpretation, and the validation of their arguments. These shortcomings necessitate a major revision of the manuscript. The severity of these issues is compounded by errors in crucial equations explaining the model. Coupled with the lack of explanation for some variables and functions, the paper in its current state poses challenges for precise interpretation. The paper would be greatly improved by revising its data presentation and argument validation, which ensure credible conclusion.

<comments methods="" on="" section="" the="">

For “Model”:

The manuscript currently lacks information for cell count and cell proliferation. It will help understanding if the authors incorporate explanations about these factors.

For “Gene expression dynamics”:

1. Equation (1) appears to be missing a ")" which compromises its correctness.

2. To enhance comprehension of the model, the authors should provide precise information about several key elements.

2-1. Equation (1) seems to describe changes in protein expression, but it remains unclear how generation is accounted for.

2-2. D_i is assumed to be a value related to diffusion, but without explanation, its exact role and method of determination remain ambiguous. This lack of clarity extends to the setting of cell-cell interactions.

2-3. The functions f(x) and F(i,x(r,t)) need proper descriptions, as their roles are currently unexplained.

2-4. The correspondence between "as x is increased" and "as x_i (l,t) decreases" needs to be clarified.

For “Initial/boundary conditions”:

1. Explanations are required to better understand this part of the manuscript

1-1. It is assumed that M represents cell numbers, but without explicit clarification, this remains uncertain.

1-2. The authors claim that gene expression levels are "spatially uniform" for genes other than gene #0 and 1. It is unclear whether this signifies identical gene expression levels across all cells.

1-3. The inheritance of gene expression levels from ancestors needs further explanation. Does it imply that the clone inherits the initial state used by the progenitor cell?

For “Selection”:

More information should be given about the variable e^(f_i ) to clarify its role and purpose.

For “Mutation”:

There appears to be no list detailing the evolvable parameters and their potential changes. Provision of this information would assist readers in comprehending the model.

For “Calculation of the similarity indicator of gene expression patterns”:

The variable σ seems to be absent in the final sentence.

For “Calibration of the timescale of slow gene”:

1. There is ambiguity on the definition of terms and the purpose of calculations, which should be resolved.

1-1. The concept of calibrating the timescale needs an explicit explanation. The authors should provide a detailed definition of timescale, clarify the purpose of its calibration, and justify the employed manipulations.

1-2. The relationship between the introduced value k and the term "timescale" remains obscure.

<comments on="" results="" section="" the="">

For paragraph 1:

1. The term "fixed terminal state" requires explicit definition to avoid ambiguity. The Methods section mentions that fitness is determined by the number of final differentiation states, which led me to an interpretation of no specific target expression pattern. clear definitions are necessary to grasp the simulation setup accurately, namely whether this understanding is incorrect (there is some target pattern), or "fixed state" refers to a "steady state".

2. Gene expression patterns appear to be changing even at the 1000th timestep, as inferred from Fig 2A. The reason behind the choice of ending "embryogenesis" at the 1000th step requires clarification.

For paragraph 2:

1. The legend for Fig. 2A states that "Spatial pattern transitions were more frequent in the early developmental time." However, spatial differentiation seems minimal until around the 100th step. It would be beneficial if the term "spatial pattern transition" is clearly defined, quantified, and compared across developmental periods.

For paragraph 3:

1. The statement "there are some phases within which patterns change only slightly" needs clarification whether it refers to subtle changes in ontogeny or phylogeny.

2. In the legend of Fig. 2B, the "middle" period is suggested as the stage of highest similarity among lineages. If it refers to the 100th step out of 1000, it seems to fall within the "early" period, thus requiring a precise definition of the "middle" period. While I comprehend that the term "middle" does not necessarily imply a literal midpoint, and the primary focus here is the hourglass-like pattern, the current manuscript would lead to confusion for readers.

3. The previous observation of the phylotypic period suggests that the conservation of this period is persistent, regardless of the evolutionary scale. Thus, if the authors claim to recapitulate an hourglass pattern in their simulation, persistent conservation at a specific period must be demonstrated.

4. Regarding Fig S1, there seems to be a mislabel on the y-axis.

5. For Fig S2, details on the number of evolutionary trials conducted, and on the probability of the hourglass-like evolutionary outcomes would enhance the paper's value.

For paragraph 4:

1. The authors claim that the “middle” period was consistently conserved between two species that diverged at the 800th generation based on one example. This assertion would be greatly strengthened by the addition of statistical data, such as the distribution of the most conserved period. This information would be particularly beneficial when discussing whether the “middle” conservation observed in this simulation persists like the phylotypic period.

For paragraph 6:

1. In Fig. 3, the authors compared cellular expression patterns between lineages representing extant species. It would be helpful to understand the similarity of developmental patterns between ancestral species (e.g., species of the 400th generation that diverged at Branch C). If the mid-conserved period in this simulation indeed represents the phylotypic period, it is expected that ancestral species also exhibit mid-phase conservation. Comparing ancestral species would also shed light on the process of establishing constraints on the conserved period.

2. The findings from Fig S3 reveal diminished conservation of the mid-phase between “distantly” related species (inculuding those 300 generations apart within the 1000-generation simulation). This might indicate that the simulation's scope extends beyond the within-phyla evolutionary scale, which is traditionally where the hourglass model is applicable. The authors should address the appropriateness of the simulation's scale.

3. The observed pattern that developmental similarity peaks at the final developmental stage among species separated by over 300 generations may imply developmental drift, as opposed to developmental constraints. The authors are recommended to share their interpretation of this observation.

For paragraph 7:

1. I encourage the authors to clarify how many evolutionary trials were executed, and within those, how frequently the results exhibited an hourglass-like pattern, and how often they demonstrated enrichment of pleiotropic genes during the conserved middle period. Statistical data on these aspects would be useful in evaluating the validity of the findings.

For paragraph 8:

1. During actual embryogenesis, developmental noise is continuously generated. However, in this simulation, noise is only introduced at the initial step. It's unclear if this methodology accurately represents real-embryogenesis and requires an additional discussion.

2. The manuscript says Fig 4 displays the variation among clones, but it appears this is represented in Fig 5.

3. The indicator for gene expression pattern similarity in this analysis is different from those utilized in other assessments of similarity. An explanation for this deviation would be helpful.

For paragraph 9:

1. The statement referring to the necessity for developmental noise to achieve robustness requires further specificity. It should clearly define the context of robustness - against what and of what? Furthermore, the corresponding setup within the simulation that reflects this requirement needs explicit clarification. There seems to be a disconnect with the setup that aims an increase in the terminal differentiated state throughout the simulation. Given this ambiguity, the rationale behind the attenuation of initial noise is not clear.

For paragraph 12:

1. A clear description of what specifically is changing in a stepwise manner in gene expression patterns would enhance understanding. Providing a quantitative evaluation, or conversely, an example of gradual change, would be beneficial.

Paragraph 13:

1. To effectively argue that a switch occurs between initially expressed genes and late-expressed genes around the bottleneck period, the authors should support their findings with statistical analysis, as opposed to merely providing an example.

2. Ref [23] investigated developmental conservation in terms of temporal variability of gene expression patterns. Their analysis highlighted larger temporal expression divergence before, rather than after, the bottleneck period. A discussion on the compatibility between the authors' findings and those from this previous study would be appreciated.

Paragraph 14:

1. In Fig 6b, the rate of gene expression change is gradual, making it difficult to classify genes as “slow”, “normal” or “fast”. The authors should define a clear and quantitative indicator.

2. The authors argue that numerous genes are regulated by slow genes. They should clarify whether this is based on solid evidence or if it is speculative.

Paragraph 15:

1. Fig 6d illustrates the correlation between the timescale of slow genes and the similarity peak. However, in substantial instances, the slow genes reach their expression change peaks after the similarity peak (provided my understanding of the “timescale” of genes is accurate). This may suggest that the data presented here doesn't fully support the causality between slow genes and developmental conservation. The authors should discuss this point.

2. The color codes in the figure legend appear to be misaligned.

<comments discussion="" on="" section="" the="">

For paragraph 2:

1. The simulation implies that substantial developmental noise is essential during evolution. Is there empirical evidence that can corroborate this assumption?

For paragraph 3:

2. The authors state that a decrease in variance due to developmental noise will lead to a reduction in variance due to genetic difference. The underlying mechanisms of this assumption remain unclear to me (Intuitively, it seems to be natural to conclude that robustness against noise would be a byproduct of robustness against genetic variations). The readability of this section could be significantly enhanced by providing further clarification on the authors' assumptions regarding "robustness."

For paragraph 4:

1. Once again, it is essential that the authors provide explicit and objective criteria for distinguishing "slow" genes.

2. The link between the expression of slow genes and the buffering of initial variations remains ambiguous. While Figure 6 indicates a correlation between the expression change in certain genes and the peak of similarity, it doesn't necessarily demonstrate a causal relationship. The authors should support their argument with analytical results, or alternatively, they should refine the manuscript to indicate that the connection is hypotherical.

For paragraph 6:

1. The discussion is intriguing on the diversification of the mid-embryonic period among lineages and the concurrent buffering of variation within a lineage. Apart from the authors' speculation, are there any additional analytical results that shed light on the process of gene repertoire alteration while maintaining slow gene sets? One of the advantages of simulation approach is its ability to trace the evolutionary process, so an analysis of this kind could add significant value to this study.

2. I would like to ascertain whether the authors are aware of the criticisms towards the methodology and conclusions of ref [41]. (Hejnol A, Dunn CW. Animal Evolution: Are Phyla Real? Curr Biol. 2016 May 23;26(10):R424-6.)

For paragraph 7:

1. In justifying their assertion that slow genes regulate many other genes, the authors cite their own previous research. I believe presenting data fromn the simulation conducted in this study would strengthen their argument.

2. The authors' proposal is intriguing to define higher taxa based on the emergence of developmental systems (gene sets) that impose constraints. I suggest that incorporating insights from the simulations about the process of emergence of such gene sets will make this paper more insightful and impactful.</comments></comments></comments>

Reviewer #3: The review is uploaded as an attachment

**Have the authors made all data and (if applicable) computational code underlying the findings in their manuscript fully available?**

Reviewer #1: Yes

Reviewer #2: Yes

Reviewer #3: Yes

PLOS authors have the option to publish the peer review history of their article (what does this mean?). If published, this will include your full peer review and any attached files.

Reviewer #1: No

Reviewer #2: No

Reviewer #3: No
---

## [Decision Letter · Decision Letter 1]

29 Jan 2024

Dear Prof. Kaneko,

We are pleased to inform you that your manuscript 'Developmental hourglass: Verification by numerical evolution and elucidation by dynamical-systems theory' has been provisionally accepted for publication in PLOS Computational Biology.

Best regards,

Philip K Maini

Academic Editor

PLOS Computational Biology

James O'Dwyer

Section Editor

PLOS Computational Biology

Reviewer's Responses to Questions

**Comments to the Authors:**

Reviewer #1: No more comments.

Reviewer #2: The authors have clearly addressed all of my questions and made the necessary revisions. I’m now convinced that this paper is sufficiently qualified for publication. This study, which dissects the relationship between developmental systems and phenotypic evolution by taking advantage of numerical simulation, will provide valuable insights to the field of evolutionary biology.

Reviewer #3: Reading the revised manuscript, I found that all my concerns as well as other reviewers' were appropriately taken care of.

Minor points

1. A left and right parenthesis do not match in the marked paragraph in P.106

2. P.121 als -> also

**Have the authors made all data and (if applicable) computational code underlying the findings in their manuscript fully available?**

Reviewer #1: Yes

Reviewer #2: Yes

Reviewer #3: **No: **Since I could not find the codes and the complete data set, the authors should put them on a public repository following the Journal policy before publication.

PLOS authors have the option to publish the peer review history of their article (what does this mean?). If published, this will include your full peer review and any attached files.

Reviewer #1: No

Reviewer #2: No

Reviewer #3: No

---

## [Editor Report · Acceptance letter]

15 Feb 2024

PCOMPBIOL-D-23-00952R1 

Developmental hourglass: Verification by numerical evolution and elucidation by dynamical-systems theory

Dear Dr Kaneko,

I am pleased to inform you that your manuscript has been formally accepted for publication in PLOS Computational Biology. Your manuscript is now with our production department and you will be notified of the publication date in due course.

With kind regards,

Anita Estes
